# M-ReDet: A mamba-based method for remote sensing ship object detection and fine-grained recognition

Xuhui Liu[1], Chi Feng[2], Shuran Zi[1], Zhengkun Qin[2]*, Qinghe Guan[3]

**1** School of Economics, Management and Law, Jilin Normal University, Siping, China, **2** School of Information Technology, Jilin Normal University, Siping, China, **3** College of Electronic and Information Engineering, Changchun University of Science and Technology, Changchun, China

* qin_zhengkun@126.com

## Abstract

Ship object detection and fine-grained recognition of remote sensing images are hot topics in remote sensing image processing, with applications in fishing vessel operation command, merchant ship navigation route planning, and other fields. In order to improve the detection accuracy for different types of remote sensing ship objects, this paper proposes a ship object perception and feature refinement method based on the improved ReDet, called Mamba-ReDet (M-ReDet). First, this paper designs a ship object fine-grained feature extraction backbone (Mamba-ReResNet, M-ReResNet), which selects and reconstructs the unique features of different types of ship objects through the Mamba's selective memory to improve the algorithm's ability to extract fine-grained features. Secondly, the M-ReDet consists of the Ship Object Perception Module (SOPM) and the Ship Feature Refinement Module (SFRM), which can extract the ship's spatial position information from the feature map, fuse different scales of spatial position information and use this information to refine the fine-grained features to improve the detection accuracy of the algorithm for different categories of ships. Finally, we use the KFIoU and Focal Loss as the regression loss and classification loss of the algorithm to improve the accuracy of the training. The experimental results show that the $mAP_{0.5}$ of the M-ReDet algorithm on the FAIR1M(ship) and DOTAv1.0 visible light (RGB) remote sensing image datasets are 43.29% and 82.09%, respectively, which is 2.78% and 3.34% higher than that of the ReDet.

## Introduction

Remote sensing ship object detection and fine-grained recognition have important research value in fishery monitoring, navigation planning, and other fields. However, problems when using optical remote sensing images for these tasks include inconsistent ship size and rotation angle, as well as complex backgrounds. Therefore, many

**Data availability statement:** The curated data and code can be accessed at the following link: https://github.com/LG973641114/M-ReDet/releases.

**Funding:** This work is supported by the Department of Science and Technology of Jilin Province, China [YDZJ202501ZYTS600]. There was no additional external funding received for this study. Funded studies: Initials of the author who received the award: Liu Xuhui Grant numbers awarded to the author: YDZJ202501ZYTS600 Full name of the funder: Jilin Provincial Department of Science and Technology URL of the funder website: http://kjt.jl.gov.cn/. The sponsors (Jilin Provincial Department of Science and Technology) provided financial support throughthegrant YDZJ202501ZYTS600, managed by Prof. Wang Jia. However, they did not participate in the study design, data collection and analysis, decision to publish, or preparation of the manuscript. Although Prof. Wang Jia oversawthe funding allocation and supported the research infrastructure, her contributions were limited to administrative andfinancial management, and she is not listed as an author in this paper.

**Competing interests:** The authors have declared that no competing interests exist.

problems must be solved when designing a remote sensing object detection algorithm that can extract the fine-grained features of ship objects and enhance spatial positioning and feature refinement capabilities.

The commonly used remote sensing object detection algorithms can be categorized into three classes: the Convolutional Neural Network (CNN), the Transformer [1], and the Mamba [2] based remote sensing object detection algorithm. Firstly, the first class of algorithms is mainly improved based on classical networks such as YOLO [3], SSD [4], Faster R-CNN [5], and ResNet [6]. In 2021, XueYang et al. proposed the GWD algorithm [7], which uses the Gaussian Wasserstein Distance to characterize the distances between rotated boxes and solve the problem of a discontinuous range of rotation angles. The same year, XueYang et al. proposed the R3Det algorithm [8], which designed a feature refinement module to obtain the object's position information and realize feature alignment. Jiaming Han et al. proposed the ReDet algorithm [9], which encodes rotational equivariant and rotational invariant features to improve the detection accuracy of remote-sensing objects. In 2022, Liping Hou et al. proposed the SASM algorithm [10], which uses two strategies, the Shape Adaptive Selection (SAS) and the Shape Adaptive Measurement (SAM), to realize the selection and evaluation of positive samples.

The size differences of remote sensing objects are significant, and the information in the background is rich. It is important to separate the unique information of the object from the background, and the Transformer can effectively solve this problem. In 2022, Li Qingyun et al. proposed the TRD algorithm [11], which reconstructed the Transformer and effectively extracted the spatial position information of the object and the correlation information between instances. In 2023, Wei Liu et al. proposed the AMTNet algorithm [12], which combines CNN and Transformer to reconstruct the backbone network. Through feature exchange, different scales of feature maps are used to improve the network's feature extraction performance for changing regions. In 2024, Mingji Yang et al. proposed the Hybrid DETR algorithm [13], which can extract alienation features between remote sensing objects and then use the alienation features to distinguish small objects in complex backgrounds through the SODM module, improving the algorithm's detection ability for small remote sensing objects.

Mamba-based remote sensing object detection algorithm is a new type of algorithm that has become more popular in recent years [14], and its selective structure reduces the computational complexity of the Transformer. It can highlight the remote sensing object's adequate feature information to improve the algorithm's robustness. In 2024, Yue Zhan et al. proposed the MambaSOD algorithm [15], which uses a dual Mamba-driven feature extractor for RGB and depth information to model remote dependencies in multimodal inputs with linear complexity. Moreover, designs a cross-modal fusion Mamba model to capture multimodal features. In the same year, Tushar Verma et al. proposed the SOAR algorithm [16], which combines the mamba and YOLOv9 [17], reduces the loss of practical information, expands the receptive field, and can effectively detect remote sensing objects.

The above three types of remote sensing rotated object detection algorithms have good detection results in their respective fields but have the following problems [18–19]:

1. Many algorithms are incompatible with different sizes of ship objects; some algorithms are only improved for small objects, and others are only improved for large and obscured objects.

2. With the deepening of the network layers, the fine-grained features of remote-sensing ship objects will be gradually lost, and some algorithms do not make long-term memory retention for the fine-grained features of remote-sensing ships.

3. The complementary information fusion between feature maps of different scales is critical, but this process also requires selective learning to reduce the interference of redundant contextual information on remote sensing ship detection and fine-grained recognition of different sizes.

To address the three problems mentioned above, this paper proposes the M-ReDet algorithm, which has the following main innovations:

1. We propose the M-Bottleneck to construct a new backbone network to achieve selective retention of fine-grained features of small ships in shallow feature maps and long-term memory of semantic features of large ships in deep feature maps and, at the same time to expand the receptive field to reduce the algorithm's false detection of objects such as berths.

2. This article designs the SFRM module to reconstruct feature maps of different resolutions and selectively supplement the information difference between feature maps of different levels to improve the algorithm's detection and fine-grained recognition capabilities for ships of different sizes.

3. This paper compares the effects of different combinations of CrossEntropyLoss, SmoothL1Loss, Focal Loss, and KFIoU used in the algorithm's training and designs several groups of comparative experiments to find the optimal loss function configuration, improve the algorithm's regression and classification accuracy, and reduce the loss of accuracy due to the imbalance in the number of different ship categories.

4. This article proposes a new remote sensing ship object detection algorithm called M-ReDet and conducts multiple comparative and ablation experiments on the FAIR1M(ship) [20] and DOTA datasets [21] to verify the effectiveness of the SOPM, SFRM modules, and optimized loss functions.

## Related work

### CNN

ReDet is a classic remote sensing object detection algorithm based on convolutional neural networks. The design of the ReDet [22] mainly aims to solve two problems. First, based on the rotation variation characteristics of convolutional neural networks, ReDet proposes a rotation-invariant backbone network to extract the rotation-invariant features of remote sensing objects. Second, the RRoIAlign only performed a spatial alignment, and there was no alignment in the channel dimension. The RiRoI Align module, which consists of RPN (Region Proposal Network) and RT (ROI Transformer), can extract the features of the rotation recommendation region for classification and regression.

The overall structure of the ReDet mainly consists of a backbone network (ReResNet50) and Neck (ReFPN) [23], which can effectively extract the rotation-invariant features of the remote-sensing object. After the remote sensing image passes through ReResNet50, the algorithm can first obtain the rotation-invariant features of the remote sensing object. In the computation of the rotational features of multiple orientations, these computations share the weights, dramatically reducing the number of computational parameters required for each rotation. The rotation-invariant features of multiple

orientations can be obtained by inputting the remote-sensing image of a fixed orientation. After that, we can fuse the feature maps of different layers in the ReFPN, and after the RiRoI Align module, the rotation-invariant features of the same remote sensing object can be extracted from the rotation-isotropic features, which contain features such as the tail and wing of an airplane, the aspect ratio of a ship, can enhance the accuracy of the remote sensing object detection. Fig 1 shows the network structure of the ReDet.

## Transformer

Two of the most widespread transformer in 2025, GQA (Group Query Attention) [24] and MLA (Multi-Head Latent Attention) [25], have driven the development of large models such as the Qwen [26] and DeepSeek [27], and have also provided CNN networks variety of optimization schemes, such as MSTrans [28], which uses Q, K, and V in GQA to reorganize the input vectors, and uses the MST module to multiple hierarchical feature maps for enhancing the model's ability to extract pixel-level features from buildings. Among them, the basis of attention in GQA is self-attention, which is to serialize data such as text or images and reconstruct the sequence by calculating the correlation between the sequence and the sequence to complete the extraction of specific object features. The computation processes of self-attention are shown below:

$$Q^i = W^q a^i \tag{1}$$

$$K^i = W^k a^i \tag{2}$$

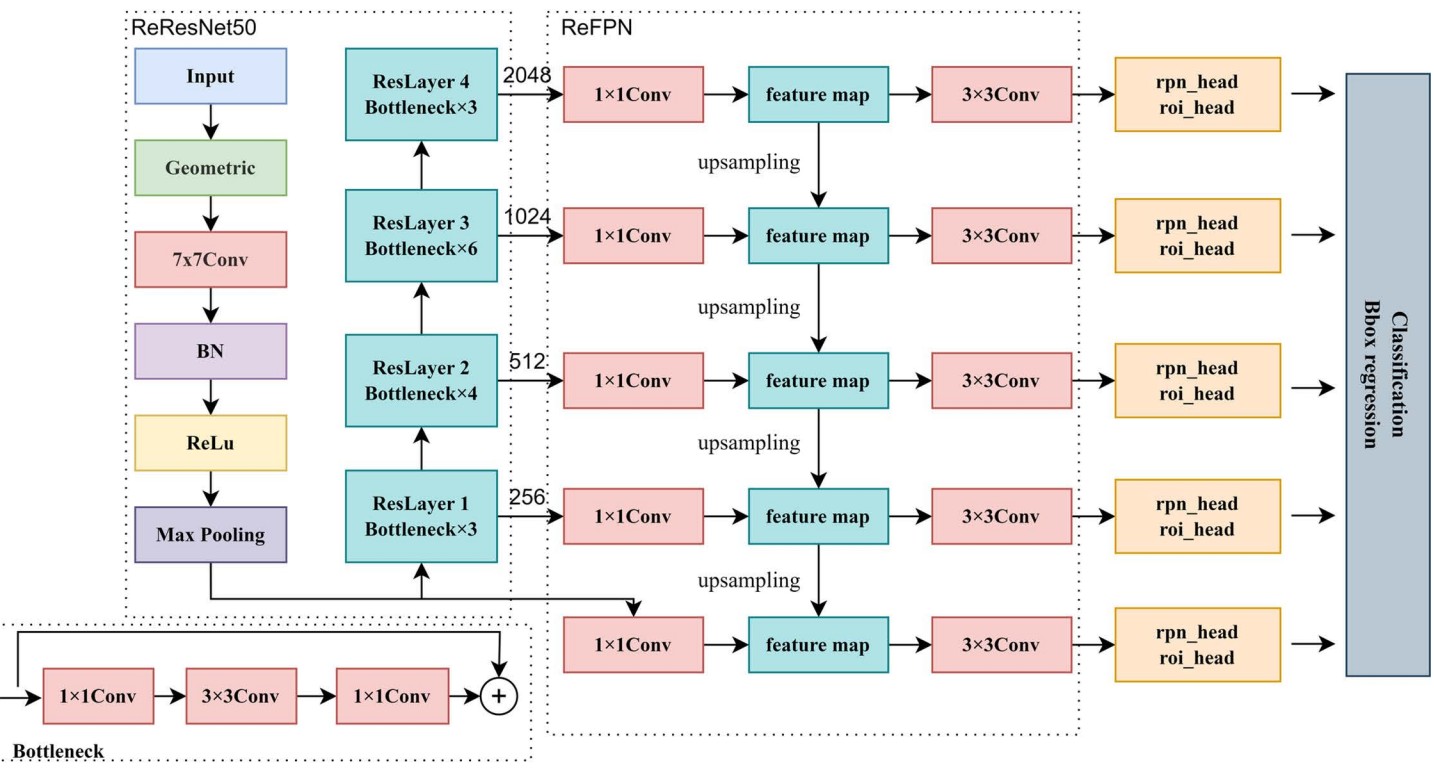

**Fig 1. Structure of the ReDet.**

$$V^i = W^v a^i \tag{3}$$

$$Attention(Q, K, V) = softmax\left(\frac{QK^T}{\sqrt{d_k}}\right) V \tag{4}$$

$a$ is the input vector, $W$ is the learnable parameter, $Q$, $K$, $V$ are the sequence computation units, and $d_k$ is the data dimension. With the increase of parallel computation and the demand to reduce the computational quantity, self-attention gradually evolves into Multi-Head Attention (MHA), Group Query Attention (GQA), and Multi-Head Latent Attention (MLA); the most important difference between them is the different methods to avoid redundant computation by KV-cache technique, the following are their computational formulas respectively:

$$y_{t;i} = softmax\left(\frac{q_{t;i}\left[k_{1;i}^T, \cdots, k_{t;i}^T\right]}{\sqrt{d}}\right) \begin{bmatrix} v_{1;i} \\ \vdots \\ v_{t;i} \end{bmatrix} \tag{5}$$

$$y_{t;i} = softmax\left(\frac{q_{t;i}\left[k_{1;g_i}^T, \cdots, k_{t;g_i}^T\right]}{\sqrt{d}}\right) \begin{bmatrix} v_{1;g_i} \\ \vdots \\ v_{t;g_i} \end{bmatrix} \tag{6}$$

$$y_{t;i} = softmax\left(\frac{q_{t;i}K_{\leq t;i}^T}{\sqrt{d}}\right) V_{\leq t;i} \tag{7}$$

$y_{t;i}$ is the output of the attention mechanism; $q$, $k$, $v$, and $d$ are the same as $Q$, $K$, $V$, and $d_k$, $g$ is the number of groups.

## Mamba

Mamba is a selective attention mechanism that can reduce the computation of the Transformer, selectively extract object features, and break through the bottleneck of the traditional model in content-aware and long-range modeling. The traditional state space model (SSM) parameters are static; Mamba introduces the gating parameter $\Delta$, which discretizes the sequence. The gating parameter of Mamba, the hidden state, and the output are calculated as follows:

$$\Delta_t = Sigmoid\left(W_\Delta z_t + b_\Delta\right) \tag{8}$$

$$\overline{A}_t = e^{\Delta_t A} \tag{9}$$

$$\overline{B}_t = (\Delta_t B) \odot z_t \tag{10}$$

$$h_t = \bar{A}_t \odot h_{t-1} + \bar{B}_t \tag{11}$$

$$y_t = C \cdot h_t + D \odot z_t \tag{12}$$

$z_t$ is the input sequence, $W_\Delta$, $b_\Delta$, $C$, and $D$ are the learnable parameters, $\bar{A}_t$ and $\bar{B}_t$ are the discretization parameters, $h_t$ is the hidden state, and $y_t$ is the output sequence. Mamba's memorability can capture the contextual information of smaller targets, and the selectivity can dynamically adjust the proportion of the tail-category features to reduce the accuracy impact brought about by long-tailed distribution.

## Our work

### M-ReDet

This paper proposes the M-ReDet algorithm based on the ReDet algorithm by improving its Backbone, Neck, and Head to improve its detection and fine-grained recognition ability for ships of all sizes. The overall framework of the M-ReDet algorithm is shown in Fig 2, which consists of the M-ReResNet50, the M-ReFPN, and the Head, respectively. The specific structure of each module after improvement will be described in detail in the subsequent subsections.

### SOPM

In this paper, the ship object perception module can better extract the edge, texture, semantic, and other features of remote sensing ship objects and improve the network's ability to detect ships and fine-grained recognition in complex sea areas such as ports. The SOPM, as the main component of the backbone, can selectively extract and highlight the features of the object's area. Its memorability can also pass the features extracted by the upper level of the feature map to the next layer so that the next layer can selectively extract the contextual information of different sizes of remote-sensing ship objects. The specific structure of SOPM is shown in Fig 3.

Before the feature map enters the SOPM, the remote sensing image passes through a 7 × 7 convolutional layer to initially extract the features such as ship texture and edges. Then the feature map enters into the multilayered ResLayer,

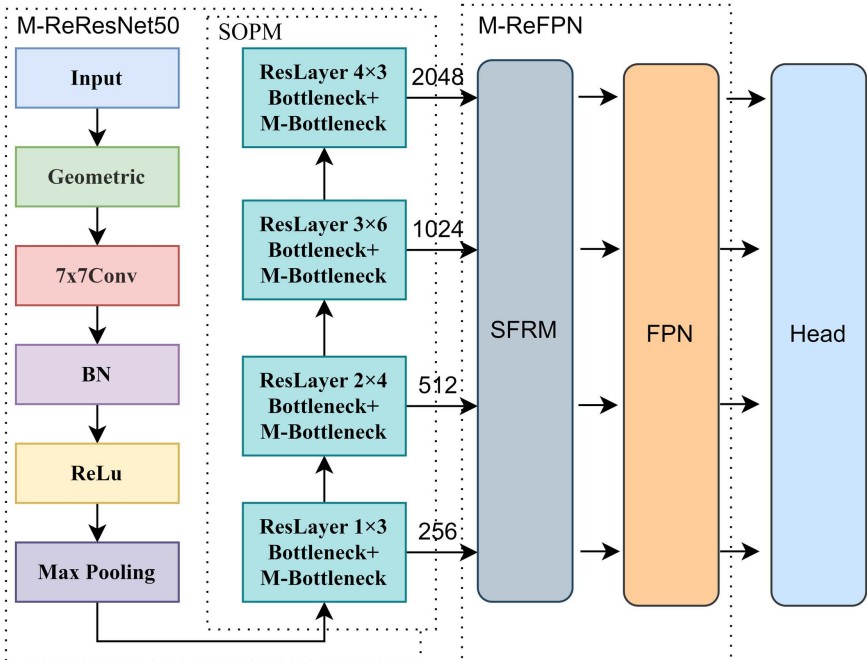

**Fig 2. The structure of the M-ReDet network.**

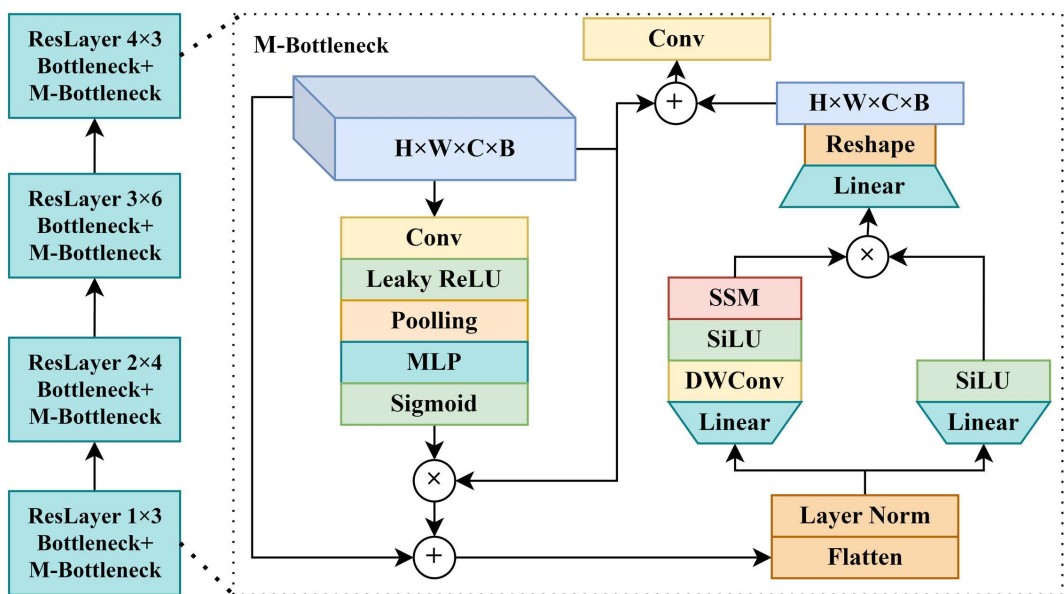

**Fig 3. The structure of the SOPM module.**

which are all composed of M-Bottleneck and Bottleneck. Inside the M-Bottleneck, the feature map is first computed by the convolution, MLP, and Sigmoid to calculate the feature map channel weights, which weight the semantic, edge, and other features of the remote sensing ships to get the fine-grained features needed for ship detection and recognition. Then, the SOPM module serializes the feature map. After the serialized features pass through the mamba's SSM module, the module can selectively extract the features of the ship object corresponding to the size of the feature map at that level and pass other features to the next level through memory to supplement the contextual information and fine-grained features of the ship corresponding to the size of the next level.

Corresponding to the four-layer ResLayer of ReResNet50, M-ReResNet50 also has four layers of ResLayers. However, the number of M-Bottlenecks in each layer differs from that of ReResNet50, and the ResLayers with different numbers of M-Bottlenecks have different capabilities for feature extraction, object perception, and localization for remote sensing ship objects. In order to find the optimal quantity ratio and ensure the total number of M-Bottleneck and Bottleneck is 3-4-6-3, this paper conducts the following experiments, setting the quantity of M-Bottleneck in each layer as 0-0-0-0, 0-1-1-0, 1-1-1-1, 1-2-2-1 and 2-2-2-2 respectively, and observing the effect of the M-Bottleneck quantity configuration on the algorithm. M-Bottleneck number configuration and the experimental results are shown in Table 1.

According to Table 1, when the number of M-Bottleneck in each layer of ResLayer is set to 1-1-1-1, the $mAP_{0.5}$ of the algorithm is the highest, which is 41.52%, and the accuracy is improved by 1.01% compared to ResLayer in ReResNet50. With the increase of the number of M-Bottlenecks, M-ReDet's accuracy does not continue to increase but stabilizes

**Table 1. Experimental results of the M-Bottleneck configuration.**

| Number | M-Bottleneck Set | mAP0.5(%) |
|---|---|---|
| 1 | 0-0-0-0 | 40.51 |
| 2 | 0-1-1-0 | 40.12 |
| 3 | 1-1-1-1 | 41.52 |
| 4 | 1-2-2-1 | 41.43 |
| 5 | 2-2-2-2 | 41.50 |

around 41.52%; adding too many M-Bottlenecks will introduce additional computation, so in this paper, we chose 1-1-1-1 as the configuration parameter of ResLayer.

## SFRM

Compared with ReResNet stacking a large number of small convolutional kernels, stacking a small number of M-Bottleneck can obtain a substantial receptive field enhancement, and theoretically, M-Bottleneck can obtain the global receptive field, which can effectively improve the algorithm's extraction ability for the contextual information of ship objects. The feature maps in the backbone network pass through the SOPM module; the algorithm can obtain the rotation-isotropic, rotation-invariant, and fine-grained features of various ship objects. The SFRM module in this section can fully use these features to fuse the upper-layer low-resolution ship fine-grained semantic features with the lower-layer high-resolution ship localization information to improve the algorithm's ship detection and fine-grained recognition accuracy. The structure diagram of the SFRM module is shown in Fig 4.

## Changing the loss function

The ReDet algorithm uses CrossEntropy Loss [29] and SmoothL1 Loss [30] as the classification and regression losses in the Head, respectively. The selection of the loss function has a particular impact on the algorithm's classification and regression accuracy. In this subsection, the classification and regression losses of the M-ReDet are modified as the Focal Loss and KFIoU Loss.

Focal Loss can deal with the long-tailed distribution problem that exists in ship object detection and fine-grained recognition tasks by adjusting the balance factor and focus factor to change the value size of the classification loss of each ship in the FAIR1M(ship) dataset; there are a total of nine categories of ships, namely Dry Cargo ship, Engineering Ship, Fishing Boat, Liquid Cargo Ship, Motorboat, Passenger Ship, Tugboat, W-ship, and other-ship, the number of ships in each category varies is shown in Fig 5.

KFIoU Loss [31] is an approximation of SkewIoU, which represents the overlapping region of the rotated boxes by Kalman filtering; it essentially calculates the overlap rate to replace the IOU without introducing additional parameters and has a complete derivation and computation for non-overlapping scenarios, which is effective and improves the accuracy in the field of rotated object detection and fine-grained recognition. Focal Loss [32] and KFIoU Loss are calculated as shown below:

$$FocalLoss(p_t) = -\alpha_t(1-p_t)^\gamma \log(p_t) \tag{13}$$

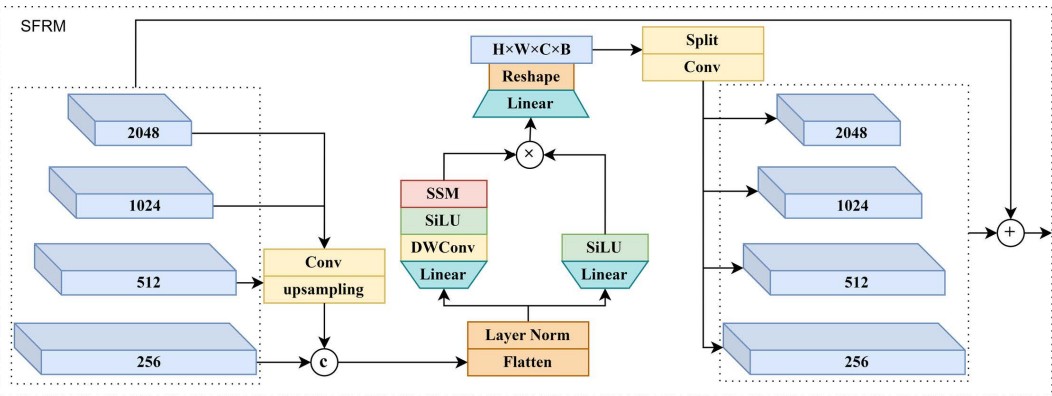

**Fig 4. Structure of the SFRM module.**

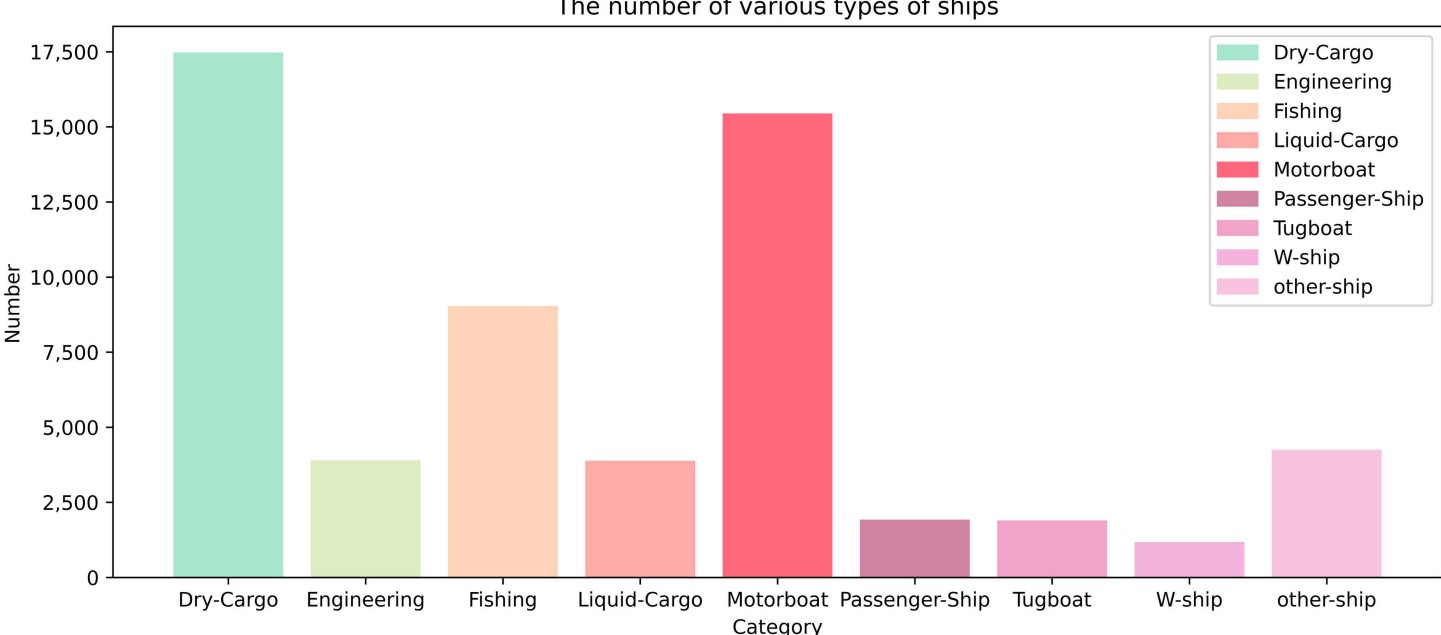

**Fig 5. The number of each type ship in the FAIR1M(ship) dataset.**

$$\mathcal{V}_{\mathcal{B}}(\Sigma) = 2^n \sqrt{\prod \mathbf{eig}(\Sigma)} = 2^n \left| \Sigma^{\frac{1}{2}} \right| = 2^n |\Sigma|^{\frac{1}{2}}$$

(14)

$$KFIoU = \frac{\mathcal{V}_{\mathcal{B}_3}(\Sigma)}{\mathcal{V}_{\mathcal{B}_1}(\Sigma_1) + \mathcal{V}_{\mathcal{B}_2}(\Sigma_2) + \mathcal{V}_{\mathcal{B}_3}(\Sigma_3)}$$

(15)

Where $p_t$ is the class label prediction probability, $\alpha_t$ is the balance factor, $\gamma$ is the focus factor, and $\alpha_t$ and $\gamma$ can adjust the effect of different categories of samples on the classification loss. $\mathcal{V}_{\mathcal{B}_1}$, $\mathcal{V}_{\mathcal{B}_2}$, and $\mathcal{V}_{\mathcal{B}_3}$ are the areas of the prediction box, the actual box, and the overlapping region, respectively. This subsection also investigates the effect of different combinations of loss functions on the accuracy of ship object detection and fine-grained recognition, and Table 2 shows the results.

As Table 2 shows, among the four loss function combinations, the mAP$_{0.5}$ of the Focal Loss + KFIoU is 41.39%, which is 0.88%, 1.07% and 0.52% higher than the CrossEntropy Loss + SmoothL1 Loss, Focal Loss + SmoothL1 Loss, and CrossEntropy Loss + KFIoU combinations, respectively. Thus, it is suitable for training the M-ReDet algorithm.

**Table 2. Experimental results for different loss functions.**

| Algorithm | Loss Set | mAP0.5(%) |
|---|---|---|
| ReDet | CrossEntropyLoss+SmoothL1Loss (original) | 40.51 |
| | Focal Loss+SmoothL1Loss | 40.32 |
| | Focal Loss+KFIoU | 41.39 |
| | CrossEntropyLoss+KFIoU | 40.87 |

## Experiment and result analysis

### Experimental environment and parameter configuration

The experimental environment used for the M-Redet remote sensing ship target detection and fine-grained recognition algorithm designed in this paper is CUDA11.1, Intel i7-11700 CPU, NVIDIA GeForce RTX 3080Ti, and the deep learning framework is PyTorch. In this paper, the comparison and ablation experiments are conducted using DOTAv1.0 and FAIR1M(ship) datasets to evaluate the detection accuracy of M-Redet and the effectiveness of each module. The experiment selects SGD as an optimizer, set lr = 0.0025, momentum = 0.9, weight_decay = 0.0001, warmup_iters = 500, warmup_ratio = 1.0/3, and the learning rate strategy is linear. The overall training has 100 epochs, and if the algorithm converges ahead, then end the training round early. Fig 6 shows the loss convergence curve of M-ReDet on the DOTAv1.0 and FAIR1M(ship) datasets. Table 3 shows the experimental environment, parameter configuration, and model resource consumption.

### Datasets

The DOTAv1.0 dataset has a total of 2806 images, which contains 15 types of remote sensing targets, namely the plane, ship, storage tank, baseball diamond, tennis court, basketball court, ground track field, harbor, bridge, large vehicle, small vehicle, helicopter, roundabout, soccer ball field, swimming pool, and the number of instances is 188282. The number of training sets in the DOTAv1.0 dataset is 1411, the number of validation sets is 458, and the number of test sets is 937. In order to facilitate the training of the algorithm, in this paper, the remote sensing images in the DOTAv1.0 dataset are cropped into 1024 × 1024 size, totaling 21046 remote sensing images. Fig 7 shows the example of the DOTAv1.0 dataset. Fig 8 illustrates the distribution of quantities for each category in the DOTAv1.0 dataset.

The FAIR1M(ship) dataset contains all remote sensing images containing remote sensing ship objects in the FAIR1M2.0 dataset. The dataset contains a total of 13238 remote sensing ship images, the number of instances is 58982, and there are a total of nine types of ship instances, namely the Dry Cargo Ship, Engineering Ship, Fishing Boat, Liquid

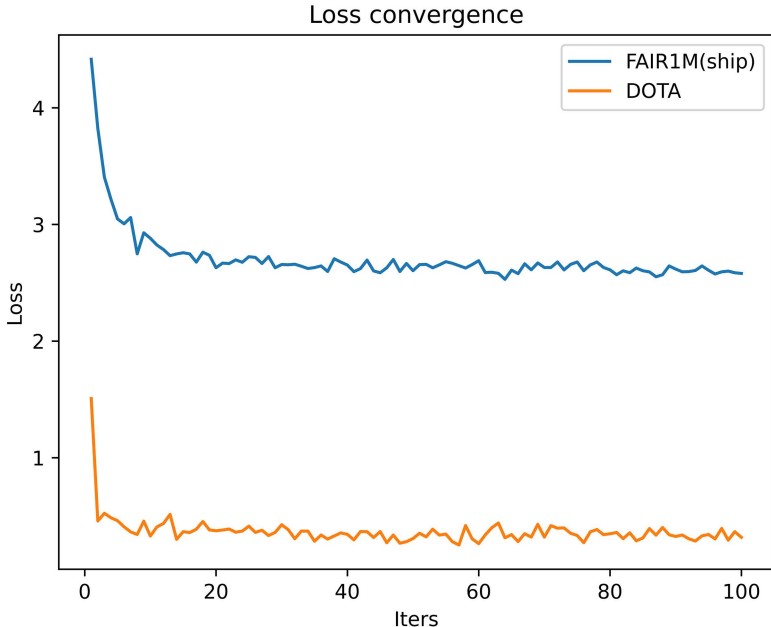

**Fig 6. Loss function curves for the training.**

**Table 3. Experimental details table.**

| Parameter | Set | Algorithm | Resource consumption |
|---|---|---|---|
| CPU | Intel i7-11700 | ReDet | Batch:5<br>GPU memory usage: 10.557G<br>Single batch time consumption: 43.8 min(8.76 min/batch) |
| GPU | RTX 3080Ti | | |
| CUDA | 11.1 | | |
| optimizer | SGD | | |
| lr | 0.0025 | | |
| momentum | 0.9 | M-ReDet (Ours) | Batch:4<br>GPU memory usage: 10.896G<br>Single batch time consumption: 31.8 min(7.95 min/batch) |
| weight_decay | 0.0001 | | |
| warmup_iters | 500 | | |
| warmup_ratio | 1.0/ 3 | | |
| DL Framework | pytorch | | |

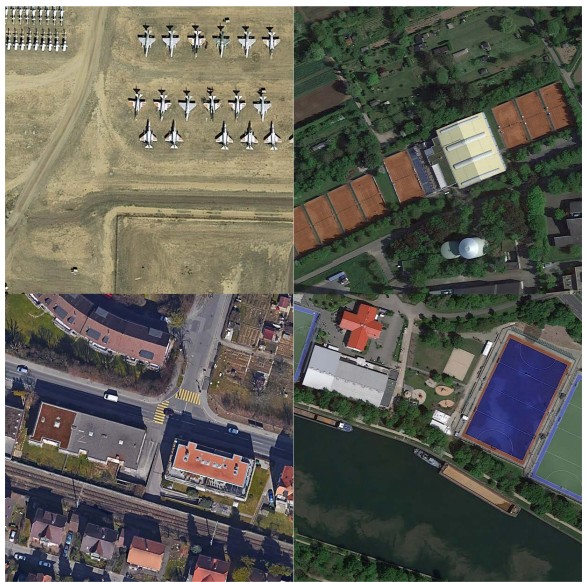

**Fig 7. Examples of the DOTA dataset.** Fig 7 is attributed to the DOTA open-source database and is available from the DOTA database (URL (s): https://captain-whu.github.io/DOTA/dataset.html).

Cargo Ship, Motorboat, Passenger Ship, Tugboat, W-ship, and other-ship. The number of ships in each category is 17474, 3897, 9031, 3883, 15445, 1924, 1897, 1178 and 4253, respectively. The FAIR1M(ship) dataset is rich in all kinds of ships, which is suitable for the research of remote sensing ship object detection and fine-grained recognition. Fig 9 shows part of the FAIR1M(ship) dataset examples.

**Experimental evaluation indicators**

We use Precision, Recall, $AP$, and $mAP$ as experimental evaluation indicators to validate the M-ReDet algorithm and the performance enhancement of each module in the comparison and ablation experiments. The formulas for Precision and Recall are as follows:

$$\text{Precision} = \frac{\text{TP}}{\text{TP} + \text{FP}} \tag{16}$$

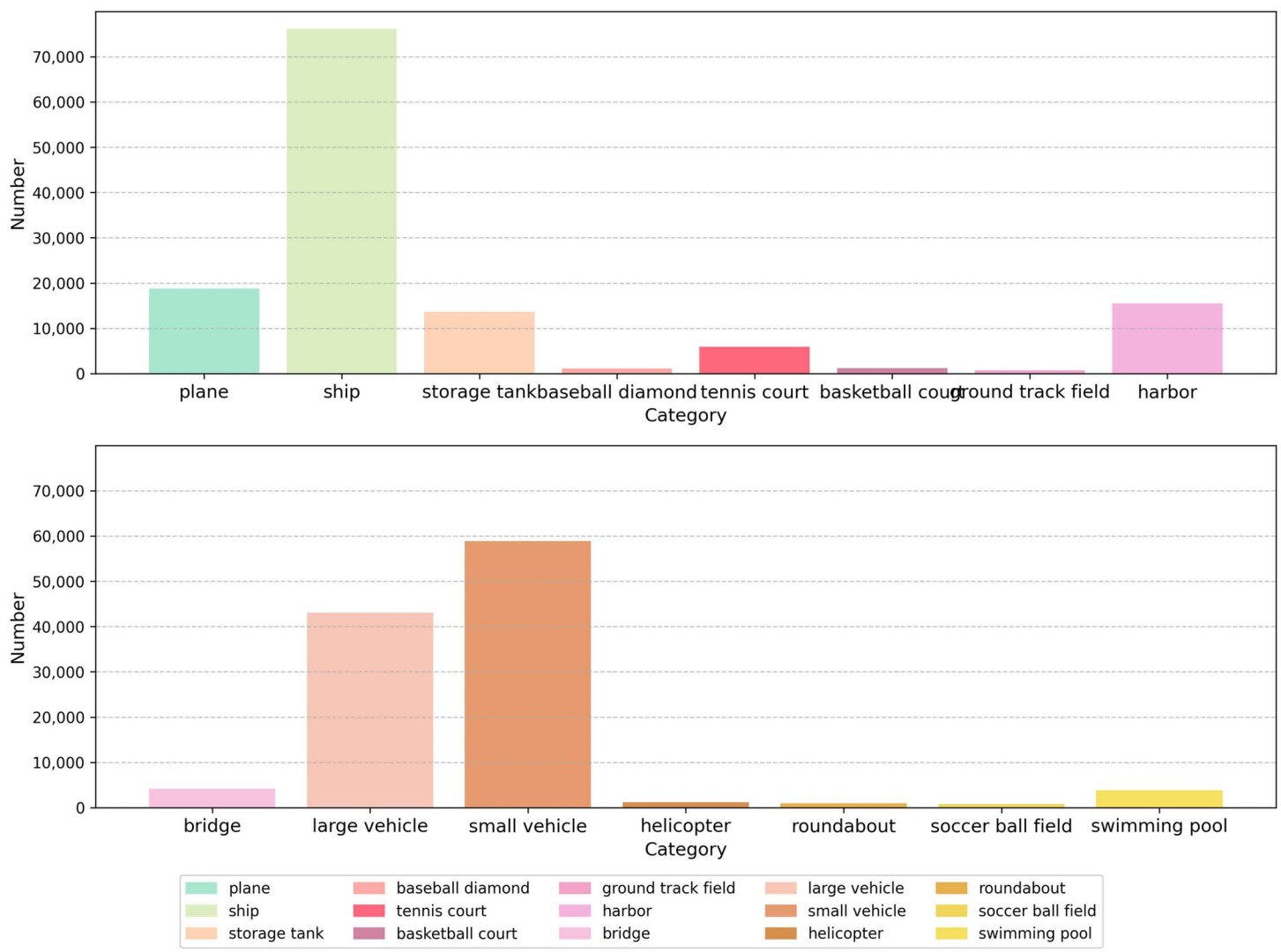

**Fig 8. The number of various types in the DOTA.**

$$\text{Recall} = \frac{TP}{TP + FN} \tag{17}$$

*TP*, *FP* and *FN* represent the number of true positives, false positives, and false negatives, respectively. Average Precision is an averaging of the accuracies at different recall rates, and in general, the model The higher the average accuracy for a specific category of target detection, the larger the AP value. The formula is as follows:

$$AP = \int_0^1 P(R)dR \tag{18}$$

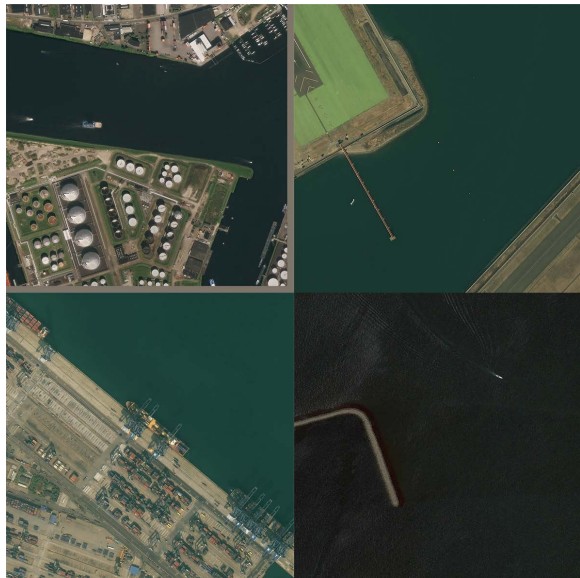

**Fig 9. Examples of the FAIR1M(ship) dataset.** Fig 9 is attributed to the FAIR1M open-source database and is available from the FAIR1M database (URL (s): https://gaofen-challenge.com/benchmark).

Where R is Recall, for a complete response to the model's overall accuracy, mAP is the average AP of all objects:

$$mAP = \frac{1}{N}\sum_{N=1}^{N} AP_i$$

(19)

N is the number of object categories, and $AP_i$ is the average accuracy of each category.

### Analysis of experimental results

**Comparison experiments on the DOTA dataset.** This subsection compares the M-ReDet algorithm with the commonly used rotated object detection algorithms such as the GWD, R³Det, RoI Transformer [33], Rotated Faster RCNN, Rotated RetinaNet, Rotated Reppoints [34], S²ANet [35], SASM, KFIoU, and ReDet to verify its effectiveness on the DOTA dataset. Fig 10 shows the detection results of the M-ReDet algorithm on the DOTA dataset.

The detection result shows that the M-ReDet can detect small, medium and large remote-sensing objects well. Faced with simple remote-sensing objects such as tennis courts and object-intensive scenarios such as harbors, the M-ReDet's detection accuracies are mostly above 90%, and some occluded and small objects have false and missed detection. Table 4 shows the experimental results of comparing the M-ReDet and the above 10 algorithms.

The AP-max of M-ReDet is similar to other algorithms but has the highest AP-min, which verifies its effectiveness in enhancing the classification and detection accuracy of remote-sensing objects with small samples. Table 5 shows the detection accuracies of various remote-sensing objects by the M-ReDet algorithm on the DOTA dataset.

The M-ReDet has the highest detection accuracy for the tennis course with an AP value of 90.8%, and for the remote sensing objects, such as planes, ships, basketball courses, harbors, large vehicles, and helicopters, also have high detection accuracies of around 90%. Since the M-ReDet uses M-Bottleneck to expand the receptive field when facing remote sensing objects with large sizes and high aspect ratios, such as bridges, the algorithms in this paper can combine the contextual information, such as selectively utilizing the road information on both sides of the bridge to make the detection

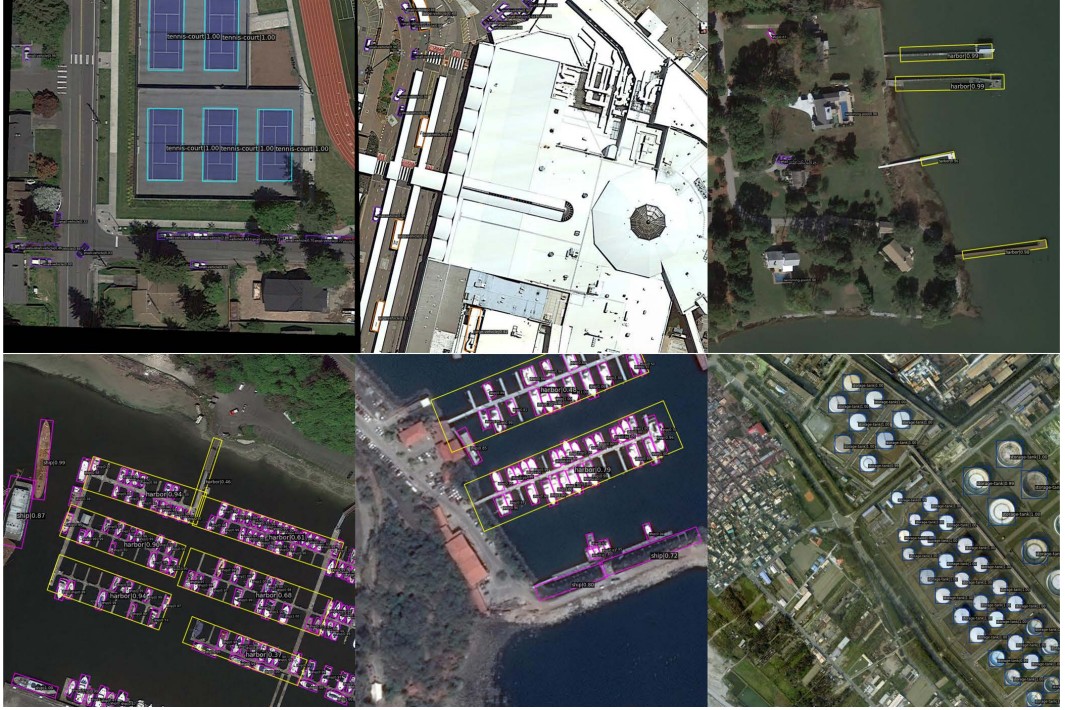

**Fig 10. Detection results of the M-ReDet on the DOTA dataset.** Fig 10 is attributed to the DOTA open-source database and is available from the DOTA database (URL **(s)**: https://captain-whu.github.io/DOTA/dataset.html).

**Table 4. Results of the DOTA dataset comparison experiment.**

| Algorithm | Size | mAP$_{0.5}$(%) | AP-max(%) | AP-min(%) | Recall-max(%) | Recall-min(%) |
|---|---|---|---|---|---|---|
| Roi-Transformer | 1024 × 1024 | 80.53* | 90.9 | 62.3 | 99.6 | 72.2 |
| SASM | 1024 × 1024 | 70.81* | 90.7 | 42.6 | 96.8 | 71.3 |
| ReDet | 1024 × 1024 | 78.75* | 90.9 | 61.3 | 97.7 | 75.5 |
| R³Det | 1024 × 1024 | 75.37* | 90.9 | 56 | 96.4 | 74.2 |
| Faster-Rcnn | 1024 × 1024 | 78.60* | 90.8 | 59.1 | 97.7 | 69.6 |
| Rotated RetinaNet | 1024 × 1024 | 77.57* | 90.8 | 52.6 | 99.2 | 73.6 |
| Rotate-reppoints | 1024 × 1024 | 66.18* | 90 | 34.4 | 96.5 | 68.9 |
| KFIoU | 1024 × 1024 | 79.88* | 90.9 | 62.6 | 99.2 | 78.3 |
| GWD | 1024 × 1024 | 79.41* | 90.9 | 62.0 | 98.9 | 83.5 |
| S²ANet. | 1024 × 1024 | 79.58* | 90.9 | 62.6 | 99.2 | 78.0 |
| M-ReDet(Ours) | 1024 × 1024 | 82.09 | 90.8 | 65.3 | 97.2 | 72.9 |

In this table,

\* means that the results from our previous work [36].

judgment. Finally, there is a 4% accuracy improvement in AP-min(bridge). Fig 11 shows the training results of the above types of algorithms.

The 11 algorithms all use the pre-trained model to train; as seen from Fig 10, the SASM and Rotated RepPoints algorithms' convergence is relatively slower compared to the other algorithms. Most of the algorithms converge to the optimal mAP$_{0.5}$ in about 10 epochs. The M-ReDet algorithm achieves convergence after 10 epochs, and the final mAP0.5

**Table 5. Detection results of various types of remote sensing objects.**

| Class | Gts | Dets | Recall(%) | AP(%) |
|---|---|---|---|---|
| plane | 4449 | 4926 | 96.1 | 90.7 |
| ship | 18537 | 21015 | 95.7 | 90.4 |
| storage tank | 4740 | 4354 | 76.4 | 72.0 |
| baseball diamond | 358 | 318 | 72.9 | 70.8 |
| tennis course | 1512 | 1654 | 96.6 | 90.8 |
| basketball course | 266 | 361 | 95.9 | 89.3 |
| ground track field | 212 | 589 | 95.3 | 83.8 |
| harbor | 4167 | 5378 | 91.6 | 87.5 |
| bridge | 785 | 1179 | 79.7 | 65.3 |
| large vehicle | 8819 | 13333 | 97.2 | 89.0 |
| small vehicle | 10579 | 15373 | 88.7 | 77.2 |
| helicopter | 122 | 175 | 96.7 | 89.5 |
| roundabout | 275 | 325 | 84.0 | 79.4 |
| soccer ball field | 251 | 495 | 93.2 | 83.9 |
| swimming pool | 732 | 1133 | 85.9 | 71.7 |

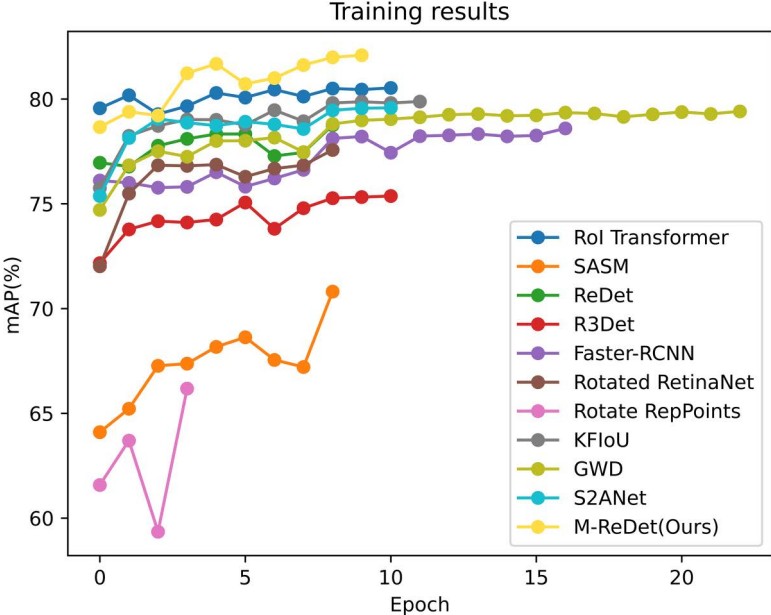

**Fig 11. Training results of each algorithm on the DOTA dataset.**

is stabilized at about 82.09%. Its training curve is also in the top left corner, indicating optimal convergence speed and accuracy.

**Comparison experiments on the FAIR1M(ship) dataset.** The M-ReDet algorithm performs well in dense ship distribution, sparse ship distribution, simple remote sensing background, and complex remote sensing background. Fig 12 shows its ship detection and fine-grained recognition results on the FAIR1M(ship) dataset. Since the appearance and aspect ratio of these nine types of ship objects are similar, it is necessary to extract the fine-grained features of each type of ship from the texture and semantic information in order to recognize the type of ship accurately. The M-ReDet algorithm

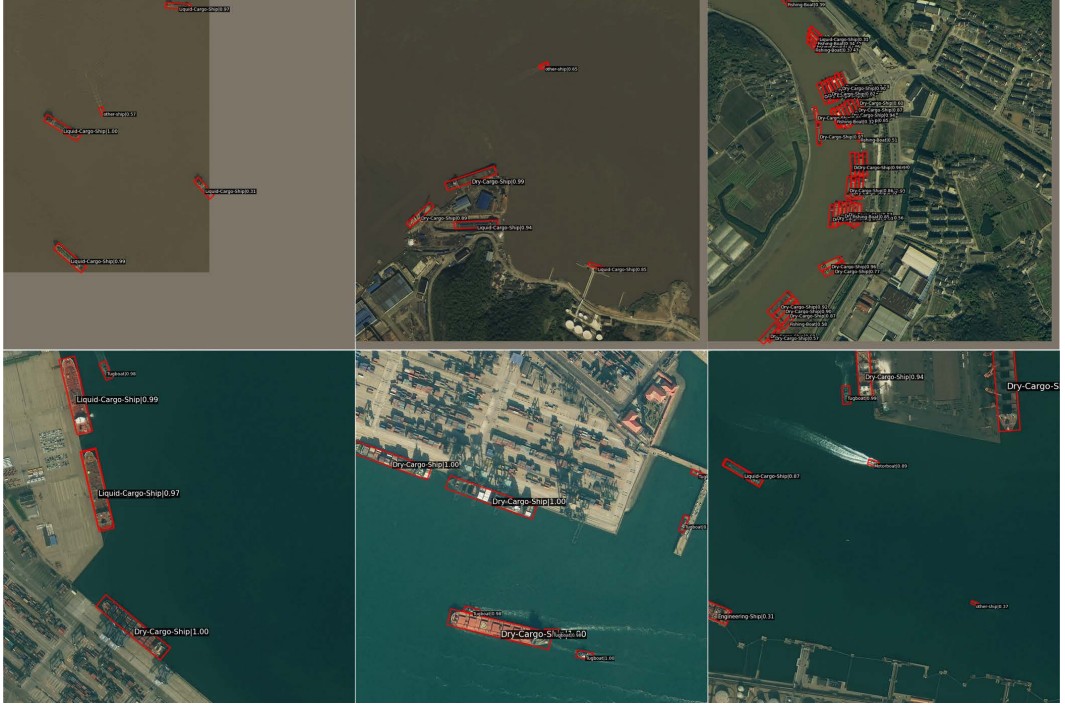

**Fig 12. Ship object detection and fine-grained recognition results.** Fig 12 is attributed to the FAIR1M open-source database and is available from the FAIR1M database (URL **(s)**: https://gaofen-challenge.com/benchmark).

first extracts the fine-grained features of the spatial location of the ship by using the SOPM and then extracts the different features between the different types of ships by using the SFRM.

In the ship detection and fine-grained recognition comparison experiments, the M-ReDet and RoI Transformer, SASM, ReDet, R³Det, Faster RCNN, Rotated RetinaNet, GWD, S²ANet, KFIoU, LSKNet [37] algorithms use the same experimental hyper-parameters and the remote sensing ship images used for training are cropped to 1024 × 1024 size. The $mAP_{0.5}$ of M-ReDet is 43.29%, which is higher than the above algorithms by 7.85%, 13.71%, 2.78%, 13.46%, 12.11%, 18.61%, 13.57%, 9.59%, 7.7% and 3.17%, respectively. M-ReDet has the highest AP-max and AP-min, which reflects from the side that the algorithm in this paper uses the SOPM and SFRM modules to enhance the size of the receptive field so that the M-ReDet can extract more contextual information, which can enhance the detection accuracy of the ship objects of all size of ships. Table 6 shows the detection results of various algorithms on the FAIR1M(ship) dataset.

There are nine categories of ship objects in the FAIR1M(ship) dataset. The detection and fine-grained recognition accuracy of ships is proportional to the number of instances of this category, except that there are many ships in the category of "other-ships" that do not have subdivided categories or the category determination information is ambiguous, which results in their detection and fine-grained recognition accuracy of only 15.9%. The M-ReDet algorithm has the highest detection accuracy for "Dry-Cargo-Ship," with an AP of 71.20%. The "Motorboat" has 7921 instances, but because of its small size, the overall AP is lower than "Dry-Cargo-Ship" at 64.7%. Table 7 shows the training results for each of the nine categories of ships.

Table 8 demonstrates the AP values of 9 types of ships detected by the 11 algorithms, respectively, to show more clearly the impact of the M-ReDet algorithm on the detection and fine-grained recognition of various types of ships. As seen from the table, the M-ReDet algorithm achieves the optimal AP in the detection results of 7 classes of ships. It is lower than the LSKNet and ReDet only in Passenger-Ship and W-ship detection results.

**Table 6. Comparison experiment results on the FAIR1M(ship) dataset.**

| Algorithm | Size | $mAP_{0.5}$(%) | $AP_{max}$(%) | $AP_{min}$(%) | $Recall_{max}$(%) | $Recall_{min}$(%) |
|---|---|---|---|---|---|---|
| RoI Transformer | 1024 × 1024 | 35.44 | 60.00 | 10.10 | 76.10 | 53.10 |
| SASM | 1024 × 1024 | 29.58 | 54.10 | 6.80 | 94.10 | 61.70 |
| ReDet | 1024 × 1024 | 40.51 | 66.30 | 10.80 | 80.50 | 54.30 |
| R³Det | 1024 × 1024 | 29.83 | 55.40 | 7.20 | 88.00 | 50.50 |
| Faster RCNN | 1024 × 1024 | 31.18 | 54.20 | 13.40 | 75.70 | 47.20 |
| Rotated RetinaNet | 1024 × 1024 | 24.68 | 44.30 | 11.00 | 89.10 | 56.20 |
| GWD | 1024 × 1024 | 29.72 | 53.50 | 6.80 | 85.90 | 48.20 |
| S₂ANet. | 1024 × 1024 | 33.70 | 60.10 | 7.10 | 91.60 | 56.20 |
| KFIoU | 1024 × 1024 | 35.59 | 60.90 | 10.60 | 76.80 | 50.90 |
| LSKNet | 1024 × 1024 | 40.12 | 66.90 | 8.50 | 87.40 | 64.80 |
| M-ReDet | 1024 × 1024 | 43.29 | 71.20 | 15.90 | 90.10 | 69.60 |

**Table 7. Training results for various types of ships.**

| Class | Gts | Dets | Recall(%) | AP(%) |
|---|---|---|---|---|
| Dry-Cargo-Ship | 8302 | 23204 | 90.10 | 71.20 |
| Engineering-Ship | 2604 | 7775 | 76.70 | 49.60 |
| Fishing-Boat | 4084 | 20058 | 77.40 | 43.20 |
| Motorboat | 7921 | 20769 | 85.00 | 64.70 |
| Liquid-Cargo-Ship | 1021 | 8704 | 69.60 | 42.40 |
| Passenger-Ship | 1391 | 6337 | 76.30 | 33.00 |
| Tugboat | 460 | 5953 | 88.00 | 30.30 |
| W-ship | 597 | 4702 | 87.60 | 39.40 |
| other-ship | 2143 | 20766 | 73.60 | 15.90 |

After 15 epochs of training, the $mAP_{0.5}$ of the M-ReDet algorithm is finally stabilized near 43.29%; because of the addition of M-Bottleneck, stabilizing the internal parameters of the module requires more training epochs of updating, and its convergence speed is slightly slower compared to the other algorithms. However, it has the highest accuracy of ship detection and fine-grained recognition, which is shown in Fig 13 for the comparison of the M-ReDet algorithm with the other 10 algorithms. Fig 13 shows the training results of all algorithms.

## Ablation experiments

The M-ReDet mainly consists of the M-ReResNet50, M-ReFPN and detection head, where the SFRM adds the M-Bottleneck and initializes it as configured in subsection 3.2. The SFRM and the FPN constitute the M-ReFPN module together. Table 9 shows the results of the ablation experiments of M-ReDet, which mainly investigate the effects of the SOPM, SFRM, KFIoU and Focal Loss alone or both or together on the $mAP_{0.5}$ of the M-ReDet algorithm, and the results of the ablation experiments are as follows.

It can be seen from Table 9 that using each module alone can improve the $mAP_{0.5}$ of the M-ReDet algorithm; the $mAP_{0.5}$ are 41.52%, 41.45%, and 41.39%, respectively, which are improved by 1.01%, 0.94%, and 0.88% compared to the base-line model. Improvement modules used two by two can also improve the $mAP_{0.5}$ of the algorithm. Finally, M-ReDet using SOPM, SFRM, KFIoU, and Focal Loss at the same time achieves the optimal ship detection and fine-grained recognition accuracy, with a mAP0.5 of 43.29%, which is 2.78% higher than that of the baseline model ReDet, and Fig 14 shows the difference between the ReDet and the M-ReDet on the FAIRM (ship) dataset for ship detection and fine-grained results.

**Table 8. Detection and fine-grained recognition results for 9 types of ships.**

| Algorithm | SASM | RoI Transformer | R³Det | Faster RCNN | Rotated RetinaNet | GWD | S²ANet. | KFIoU | LSKNet | ReDet | M-ReDet |
|---|---|---|---|---|---|---|---|---|---|---|---|
| Dry-Cargo-Ship | 54.10 | 60.00 | 55.40 | 54.20 | 44.30 | 53.50 | 60.10 | 60.90 | 66.90 | 66.30 | 71.20 |
| Engineering-Ship | 31.70 | 40.60 | 35.00 | 30.10 | 36.50 | 34.40 | 34.10 | 36.90 | 42.00 | 46.40 | 49.60 |
| Fishing-Boat | 21.40 | 28.70 | 22.30 | 19.00 | 21.60 | 20.60 | 27.70 | 29.30 | 39.40 | 38.20 | 43.20 |
| Motorboat | 42.80 | 54.10 | 38.30 | 46.40 | 32.00 | 36.40 | 44.00 | 55.20 | 60.20 | 56.80 | 64.70 |
| Liquid-Cargo-Ship | 34.00 | 34.00 | 31.30 | 30.50 | 17.60 | 33.10 | 35.60 | 34.60 | 39.60 | 37.80 | 42.40 |
| Passenger-Ship | 23.10 | 25.10 | 26.80 | 28.60 | 20.30 | 24.30 | 27.10 | 29.70 | 35.90 | 34.90 | 33.00 |
| Tugboat | 21.60 | 27.50 | 25.80 | 27.70 | 25.30 | 25.90 | 27.30 | 28.70 | 26.40 | 28.30 | 30.30 |
| W-ship | 30.80 | 38.90 | 26.40 | 30.60 | 11.00 | 32.50 | 40.30 | 34.40 | 42.10 | 45.20 | 39.40 |
| other-ship | 6.80 | 10.10 | 7.20 | 13.40 | 13.50 | 6.80 | 7.10 | 10.60 | 8.50 | 10.80 | 15.90 |
| mAP(%) | 29.58 | 35.44 | 29.83 | 31.18 | 24.68 | 29.72 | 33.70 | 35.59 | 40.12 | 40.51 | 43.29 |

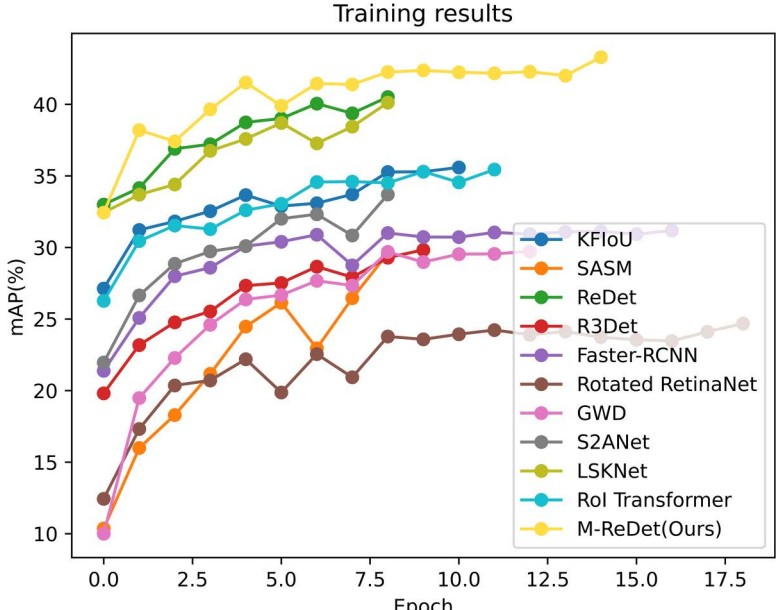

**Fig 13. Comparison of training results on the FAIR1M(ship) dataset.**

**Table 9. Ablation experiments of the M-ReDet.**

| Algorithm | SOPM | SFRM | KFIoU+Focal | mAP$_{0.5}$(%) |
|---|---|---|---|---|
| ReDet | | | | 40.51 |
| ReDet+SOPM | √ | | | 41.52 |
| ReDet+SFRM | | √ | | 41.45 |
| ReDet+KFIoU+Focal | | | √ | 41.39 |
| ReDet+SOPM+KFIoU+Focal | √ | | √ | 42.25 |
| ReDet+SOPM+SFRM | √ | √ | | 42.60 |
| ReDet+SFRM+KFIoU+Focal | | √ | √ | 42.00 |
| M-ReDet(ours) | √ | √ | √ | 43.29 |

From [Fig 14](), it can be found that the SOPM module in the M-ReDet algorithm expands the receptive field, which enables the algorithm to extract more contextual information, such as the sea surface and ports. With that information, the M-ReDet algorithm can detect the remote sensing ship objects that are in the edge position or obscured; for example, in the first line of the remote sensing image, M-ReDet successfully detects the incomplete ship objects that are on the top, right and bottom left, respectively. Moreover, the SOPM and SFRM modules can selectively memorize the fine-grained features of remote sensing ship objects of different sizes to minimize the loss of features of small ship objects in the downsampling process and to improve the algorithm's ability to detect small ships, for example, the first image in the second row, M-ReDet detects a tiny "other-ship" object located in the center of the image, but its size is too small, and the classification confidence is low. In addition, the enlarged receptive field of M-ReDet can also avoid the false detection of large buildings such as bridges and dock berths; for example, in the second row, ReDet recognizes the dock berth in the second image and the bridge in the third image as a ship.

## Discussion

The main innovation of this paper is the design of the M-ReDet algorithm framework. The proposal of two improved modules, SOPM and SFRM, which form a new backbone network (M-ReResNet50) and a new Neck (M-ReFPN), which enable the algorithm to selectively learn and retain fine-grained information about objects of different sizes of ships, and to fuse the contextual information required for memorization, which improves the algorithm's detection and fine-grained recognition ability for ships of different size. After several sets of comparison and ablation experiments, the detection robustness of the M-ReDet algorithm designed in this paper is optimal.

Small ship feature information is easily lost in the downsampling process, resulting in the loss of detection and fine-grained recognition accuracy. The M-Bottleneck in the SOPM module can extract and memorize the fine-grained features of small ships and send them to the subsequent layers for further feature extraction. In the SFRM module, the feature maps of different layers can also realize the complementary selective information so that the information on ships of different sizes can be better retained. The SOPM module can also expand the receptive field and cooperate with the SFRM module to supplement the contextual information required by different scale feature maps to reduce the probability of misdetection of similar ship objects. The optimization of classification loss and regression loss is also essential for the algorithm training process. However, this paper selects only four commonly used loss functions for permutation and combination to try the optimal loss function scheme. There are no further attempts to compare and analyze the recent excellent loss functions. Due to the limitation of hardware memory, this paper also did not further improve the number of configurations of M-Bottleneck in the backbone network to optimize the module structure and training hyperparameters of M-Bottleneck based on the experimental training results. In the future, we will make improvements to the structure and configuration of M-Bottleneck in subsection 3.2, continue to optimize the training loss function of the algorithm, and try to incorporate the fine-grained feature information into the loss function for the ship detection and fine-grained recognition tasks further to improve the regression and classification accuracy of the algorithm. In addition, this study can be integrated with other remote sensing data in the future, allowing for the design of multimodal remote sensing image fusion modules that combine visible light, SAR, and infrared data to enhance the algorithm's remote sensing ship detection capability in harsh weather conditions, such as cloudy and foggy scenes; other fields can also utilize this work, such as agricultural object detection [38–39].

## Conclusion

Remote sensing ship detection and fine-grained recognition face significant challenges due to high inter-class similarity in aspect ratios, ambiguous appearance features among vessel categories, arbitrary orientation variations, and multi-scale object characteristics. This paper proposes the M-ReDet, a memory-augmented ship perception network with feature refinement mechanisms, to address these issues. The optimization of the loss function of M-ReDet further improves the

**A**

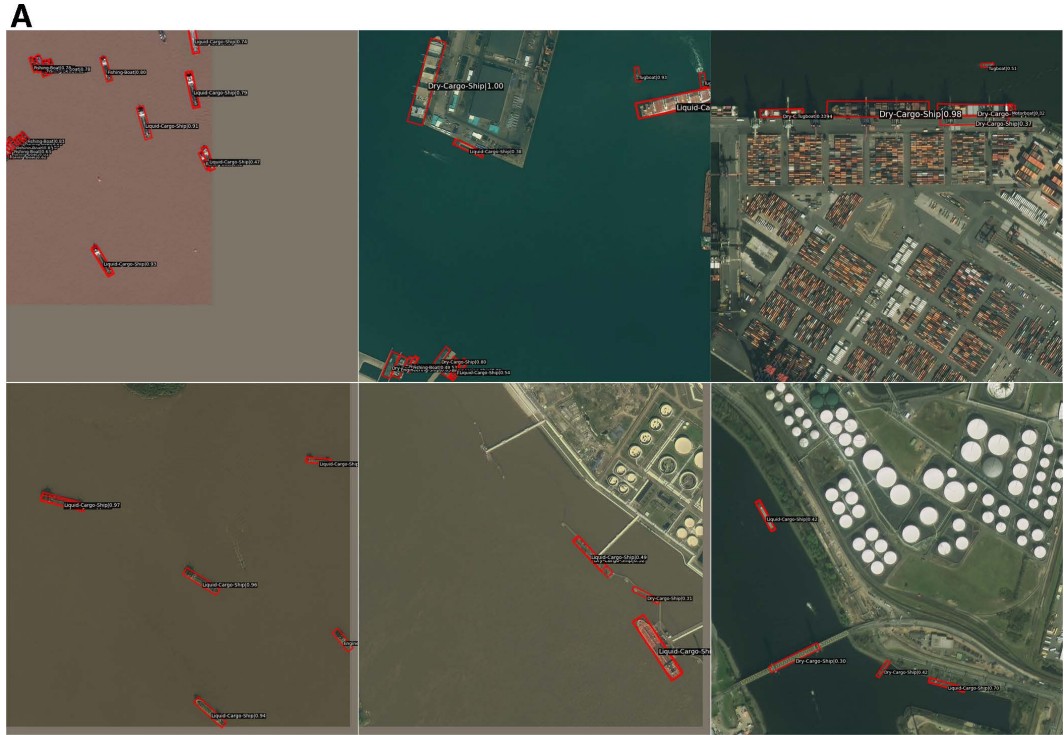

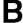

**B**

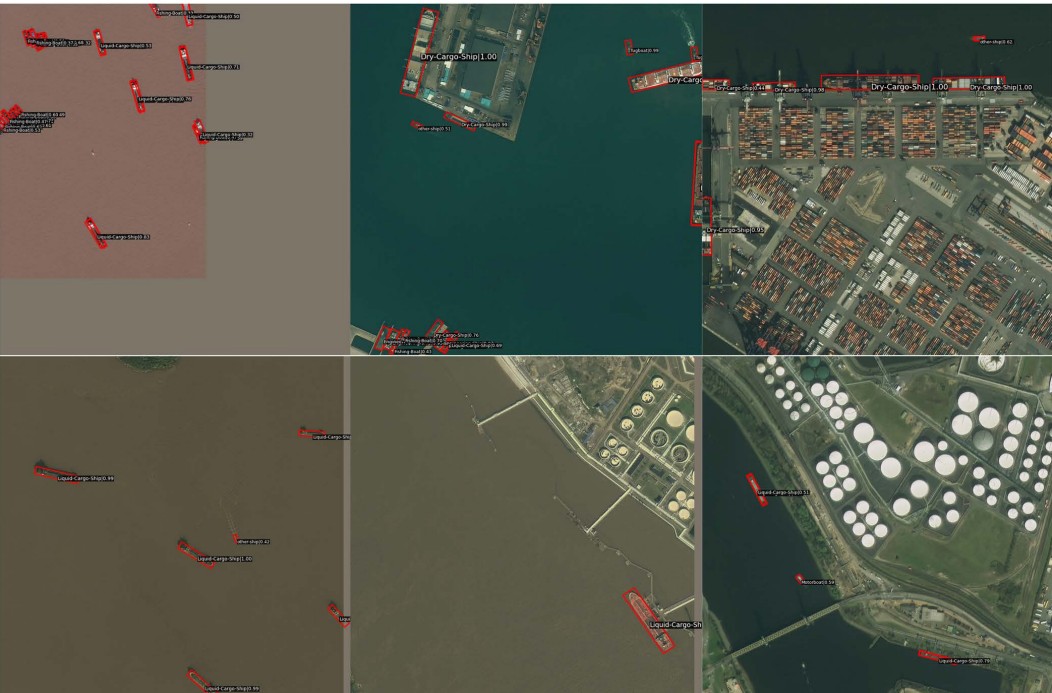

**Fig 14. Ship detection and fine-grained results.** **(a)**Detection and fine-grained results of the ReDet; **(b)**Detection and fine-grained results of the M-ReDet. Fig 14 is attributed to the FAIR1M open-source database and is available from the FAIR1M database (URL (s): https://gaofen-challenge.com/benchmark).

classification and regression accuracy of the algorithm. The comparative and ablation experiments on the FAIRM(ship) and DOTA datasets prove the effectiveness of this paper's improved modules. Finally, in the remote sensing ship and fine-grained recognition task, the M-ReDet's $mAP_{0.5}$ is 43.29%, validating the effectiveness of our algorithm in complex maritime scenarios.

## Acknowledgments

The author would like to express thanks to anonymous reviewers for all careful review of the paper and kind suggestions made to improve overall quality of the manuscript.

## Author contributions

**Conceptualization:** Xuhui Liu, Chi Feng.

**Data curation:** Xuhui Liu, Shuran Zi.

**Funding acquisition:** Xuhui Liu.

**Methodology:** Xuhui Liu, Chi Feng.

**Software:** Xuhui Liu, Chi Feng, Qinghe Guan.

**Supervision:** Zhengkun Qin.

**Writing – original draft:** Xuhui Liu, Chi Feng, Shuran Zi, Qinghe Guan.

**Writing – review & editing:** Shuran Zi, Zhengkun Qin, Qinghe Guan.

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
