## [Decision Letter · Decision Letter 0]

4 Jul 2025

PONE-D-25-19846M-ReDet: A Mamba-based Method for Remote Sensing Ship Object Detection and Fine-grained RecognitionPLOS ONE

Dear Dr. Qin,

Thank you for submitting your manuscript to PLOS ONE. After careful consideration, we feel that it has merit but does not fully meet PLOS ONE’s publication criteria as it currently stands. Therefore, we invite you to submit a revised version of the manuscript that addresses the points raised during the review process.

**Revise the paper based on referee comments.** Please submit your revised manuscript by Aug 18 2025 11:59PM. If you will need more time than this to complete your revisions, please reply to this message or contact the journal office at plosone@plos.org . Please include the following items when submitting your revised manuscript:

We look forward to receiving your revised manuscript.

Kind regards,

Fatih Uysal, Ph.D.

Academic Editor

PLOS ONE

**Journal Requirements:**

1. When submitting your revision, we need you to address these additional requirements. Please ensure that your manuscript meets PLOS ONE's style requirements, including those for file naming. The PLOS ONE style templates can be found at https://journals.plos.org/plosone/s/file?id=wjVg/PLOSOne_formatting_sample_main_body.pdf and https://journals.plos.org/plosone/s/file?id=ba62/PLOSOne_formatting_sample_title_authors_affiliations.pdf 2. Please note that PLOS ONE has specific guidelines on code sharing for submissions in which author-generated code underpins the findings in the manuscript. In these cases, we expect all author-generated code to be made available without restrictions upon publication of the work. Please review our guidelines at https://journals.plos.org/plosone/s/materials-and-software-sharing#loc-sharing-code and ensure that your code is shared in a way that follows best practice and facilitates reproducibility and reuse. 3. Thank you for stating in your Funding Statement: This work is supported by the Department of Science and Technology of Jilin Province, China [YDZJ202501ZYTS600].Funded studies:Initials of the author who received the award: Liu XuhuiGrant numbers awarded to the author: YDZJ202501ZYTS600Full name of the funder: Jilin Provincial Department of Science and TechnologyURL of the funder website: http://kjt.jl.gov.cn/The sponsors (Jilin Provincial Department of Science and Technology) provided financial support through the grant YDZJ202501ZYTS600, managed by Prof. Wang Jia. However, they did not participate in the study design, data collection and analysis, decision to publish, or preparation of the manuscript. Although Prof. Wang Jia oversaw the funding allocation and supported the research infrastructure, her contributions were limited to administrative and financial management, and she is not listed as an author in this paper.  Please provide an amended statement that declares *all* the funding or sources of support (whether external or internal to your organization) received during this study, as detailed online in our guide for authors at http://journals.plos.org/plosone/s/submit-now.  Please also include the statement “There was no additional external funding received for this study.” in your updated Funding Statement. Please include your amended Funding Statement within your cover letter. We will change the online submission form on your behalf. 4. Thank you for uploading your study's underlying data set. Unfortunately, the repository you have noted in your Data Availability statement does not qualify as an acceptable data repository according to PLOS's standards. At this time, please upload the minimal data set necessary to replicate your study's findings to a stable, public repository (such as figshare or Dryad) and provide us with the relevant URLs, DOIs, or accession numbers that may be used to access these data. For a list of recommended repositories and additional information on PLOS standards for data deposition, please see https://journals.plos.org/plosone/s/recommended-repositories. 5. We note that Figures 7, 8, 9, 11, 13a and 13b in your submission contain satellite images which may be copyrighted. All PLOS content is published under the Creative Commons Attribution License (CC BY 4.0), which means that the manuscript, images, and Supporting Information files will be freely available online, and any third party is permitted to access, download, copy, distribute, and use these materials in any way, even commercially, with proper attribution. For these reasons, we cannot publish previously copyrighted maps or satellite images created using proprietary data, such as Google software (Google Maps, Street View, and Earth). For more information, see our copyright guidelines: http://journals.plos.org/plosone/s/licenses-and-copyright. We require you to either present written permission from the copyright holder to publish these figures specifically under the CC BY 4.0 license, or remove the figures from your submission: a. You may seek permission from the original copyright holder of Figures 7, 8, 9, 11, 13a and 13b to publish the content specifically under the CC BY 4.0 license.   We recommend that you contact the original copyright holder with the Content Permission Form (http://journals.plos.org/plosone/s/file?id=7c09/content-permission-form.pdf) and the following text:“I request permission for the open-access journal PLOS ONE to publish XXX under the Creative Commons Attribution License (CCAL) CC BY 4.0 (http://creativecommons.org/licenses/by/4.0/). Please be aware that this license allows unrestricted use and distribution, even commercially, by third parties. Please reply and provide explicit written permission to publish XXX under a CC BY license and complete the attached form.” Please upload the completed Content Permission Form or other proof of granted permissions as an "Other" file with your submission. In the figure caption of the copyrighted figure, please include the following text: “Reprinted from [ref] under a CC BY license, with permission from [name of publisher], original copyright [original copyright year].” b. If you are unable to obtain permission from the original copyright holder to publish these figures under the CC BY 4.0 license or if the copyright holder’s requirements are incompatible with the CC BY 4.0 license, please either i) remove the figure or ii) supply a replacement figure that complies with the CC BY 4.0 license. Please check copyright information on all replacement figures and update the figure caption with source information. If applicable, please specify in the figure caption text when a figure is similar but not identical to the original image and is therefore for illustrative purposes only.The following resources for replacing copyrighted map figures may be helpful: USGS National Map Viewer (public domain): http://viewer.nationalmap.gov/viewer/The Gateway to Astronaut Photography of Earth (public domain): http://eol.jsc.nasa.gov/sseop/clickmap/Maps at the CIA (public domain): https://www.cia.gov/library/publications/the-world-factbook/index.html and https://www.cia.gov/library/publications/cia-maps-publications/index.htmlNASA Earth Observatory (public domain): http://earthobservatory.nasa.gov/Landsat:
http://landsat.visibleearth.nasa.gov/USGS EROS (Earth Resources Observatory and Science (EROS) Center) (public domain): http://eros.usgs.gov/#Natural Earth (public domain): http://www.naturalearthdata.com/

**Additional Editor Comments:**

Revise the paper based on referee comments.

Reviewers' comments:

Reviewer's Responses to Questions

**Comments to the Author**

1. Is the manuscript technically sound, and do the data support the conclusions?

Reviewer #1: Yes

Reviewer #2: Yes

2. Has the statistical analysis been performed appropriately and rigorously? 

Reviewer #1: Yes

Reviewer #2: Yes

3. Have the authors made all data underlying the findings in their manuscript fully available?

Reviewer #1: Yes

Reviewer #2: Yes

4. Is the manuscript presented in an intelligible fashion and written in standard English?

Reviewer #1: Yes

Reviewer #2: Yes

5. Review Comments to the Author

**Reviewer #1: ** The work is well organised and successfully presented. The following paper sets out a proposal for a new detection algorithm with regard to the identification of ships. It is hypothesised that the study will achieve a higher level of success if the following recommendations are taken into consideration during the process of its update.

1. It is important to note that all tables and graphs in the study were prepared with the mAP metric exclusively. The application of additional overlap metrics is also recommended. It is recommended that metrics such as DSC, IoU,RAVD and HD95 be deployed.

2. It is imperative that references are updated and expanded.

3. It is recommended that the resolution quality of the images is increased, with a view to enhancing their clearness and legibility.

**Reviewer #2:**  I appreciate the effort put into this manuscript. Addressing the suggested revisions will significantly enhance the clarity and impact of your work. By refining these sections, the manuscript will provide greater value to the research community and result in a more robust and informative publication.

6. PLOS authors have the option to publish the peer review history of their article (what does this mean? ). If published, this will include your full peer review and any attached files.

**Do you want your identity to be public for this peer review?** For information about this choice, including consent withdrawal, please see our Privacy Policy .

Reviewer #1: **Yes: ** Mehmet Süleyman YILDIRIM

Reviewer #2: **Yes: ** Ozan PEKER

---

## [Author Response · Author response to Decision Letter 1]

11 Jul 2025

Response to the Reviewers’ Comments

Thank you very much for giving us the opportunity to improve the manuscript. The authors are grateful for the reviewers’ valuable comments, which have helped us to understand the underlying approach in depth and improved the original manuscript. We think carefully about comments raised by the reviewers and provide an item-by-item list to explain how these comments were addressed.

In the 'Revised Manuscript with Track Changes' file, we use the highlighting to mark the modifications. Point-to-point responses to reviewers’ comments are below. Reviewers’ comments are in italics and blue, immediately followed by our response.

Reviewer: 1

Comments to the Author:

The work is well organised and successfully presented. The following paper sets out a proposal for a new detection algorithm with regard to the identification of ships. It is hypothesised that the study will achieve a higher level of success if the following recommendations are taken into consideration during the process of its update.

Response Thank you very much for your careful review and recognition of our manuscript. We will make cautious revisions based on your feedback. Thank you very much.

Specific comments:

1. It is important to note that all tables and graphs in the study were prepared with the mAP metric exclusively. The application of additional overlap metrics is also recommended. It is recommended that metrics such as DSC, IoU,RAVD and HD95 be deployed.

Response Thank you very much for your valuable feedback. The mAP is critical in remote sensing object detection tasks. Additionally, the evaluation metrics in the manuscript also utilize AP and Recall. In the loss function, we also use IoU as the training basis for the algorithm. The DSC and other metrics you mentioned are used mainly in the field of image segmentation. Thank you again for your valuable feedback.

2. It is imperative that references are updated and expanded.

Response Thank you very much for raising the questions in the references. We have revised the format of the references and added some references.

3.It is recommended that the resolution quality of the images is increased, with a view to enhancing their clearness and legibility.

Response Thank you very much for your valuable feedback. Our minimum image resolution is 1024 × 1024, and the image displayed in the PDF file should be a resized version of the original image. You can download and view the original image in the system. Thank you very much for your efforts and recognition.

Reviewer: 2

Comments to the Author:

I appreciate the effort put into this manuscript. Addressing the suggested revisions will significantly enhance the clarity and impact of your work. By refining these sections, the manuscript will provide greater value to the research community and result in a more robust and informative publication.

Response Thank you very much for giving us the opportunity to improve the manuscript and pointing out the shortcomings in the manuscript. In this iteration, we have revised the manuscript on the following issues.

Specific comments:

1. Abstract: In addition to emphasizing that ship detection in remote sensing imagery is currently a hot topic, it would be beneficial to specify the application domains in which such detection tasks are conducted. Doing so would help the reader better understand the practical relevance of the research question and potentially reveal its target users or beneficiaries. Please specify the image format (RGB,Multispectral,etc.) of the dataset used in the study within the abstract. This information will enhance the clarity and reproducibility of the work.

Response Thank you very much for your valuable feedback. Adding application areas and image formats to the abstract can indeed improve the clarity of the research. We have added relevant content.

2. Introduction: Please clarify the novelty of the study more explicitly at the end of the Introduction section. Clearly stating the unique contributions will help distinguish this work from existing literature.

Response Thank you for your efforts during the review process. We have made revisions to the work description and innovative design at the end of the manuscript introduction section.

3. Related work: The content of the Related Work section is expected to align with the statement in your sentence: "The commonly used remote sensing object detection algorithms can be categorized into three classes: the Convolutional Neural Network, the Transformer [1], and the Mamba...". Ensuring coherence between this categorization and the subsequent discussion will improve the clarity and structure of the manuscript.

Response Thank you very much for your careful review. We have revised the titles of the relevant work sections to align them with the descriptions in the introduction section of the manuscript.

4. Our Work: Under the heading "Optimizing the Loss Function," merely changing the loss function appears inconsistent. Readers may expect an actual optimization method or strategy within this section. It is advisable to either revise the heading to better reflect the content or expand the section to include optimization techniques. In addition to KFIoU Loss, it is recommended to include a information about Focal Loss as well.

Response We are very grateful for your valuable suggestions. We have revised the title "Optimizing the Loss Function" to "Changing the Loss Function" to avoid confusion for readers. The manuscript also includes a description of Focal loss.

5. Experiment and Result Analysis: Presenting the details under the "Experimental Environment and Parameter Configuration" section in a tabular format would enhance readability and clarity for the reader. Additionally, it is recommended to include resource consumption metrics of your developed model compared to the baseline model, such as RAM usage and training time per epoch. These details will provide a more comprehensive evaluation of the model’s efficiency. Please provide detailed visualizations of the distribution within the datasets using appropriate graphs. This will help readers better understand the data composition and support the validity of the experimental results.

Response Thank you for your question regarding the experimental configuration and dataset situation. We have added Table 3, which outlines the parameters configuration and resource consumption, and Fig 8 to illustrate the dataset's distribution.

6. Discussion: In the Discussion section, it would be beneficial to mention potential future work, such as leveraging SAR imagery for improved performance and addressing enhancements to cope with adverse weather conditions. Including these aspects could strengthen the manuscript by highlighting promising directions for further research.

Response Thank you for your suggestion about future work in the discussion section. Indeed, adding possible future application directions and plans can provide readers with more inspiration. We have added some potential future work and application directions in the discussion section.

---

## [Editor Report · Decision Letter 1]

4 Aug 2025

M-ReDet: A Mamba-based Method for Remote Sensing Ship Object Detection and Fine-grained Recognition

PONE-D-25-19846R1

Dear Dr. Qin,

We’re pleased to inform you that your manuscript has been judged scientifically suitable for publication and will be formally accepted for publication once it meets all outstanding technical requirements.

Kind regards,

Fatih Uysal, Ph.D.

Academic Editor

PLOS ONE

Additional Editor Comments (optional):

Following a thorough evaluation of the final version of the paper, the responses to the reviewer comments, and the revisions implemented, the manuscript was accepted based on its potential contribution to the literature.
---

## [Editor Report · Acceptance letter]

PONE-D-25-19846R1

PLOS ONE

Dear Dr. Qin,

I'm pleased to inform you that your manuscript has been deemed suitable for publication in PLOS ONE. Congratulations! Your manuscript is now being handed over to our production team.

Kind regards,

on behalf of

Assoc. Prof. Dr. Fatih Uysal

Academic Editor

PLOS ONE